# Large-Scale Similarity Search with Optimal Transport

**Cléa Laouar**[1], **Yuki Takezawa**[1,2], **Makoto Yamada**[1]

[1]Okinawa Institute of Science and Technology, [2]Kyoto University

c.laouar@oist.jp, yuki-takezawa@ml.ist.i.kyot-u.ac.jp, makoto.yamada@oist.jp

## Abstract

Wasserstein distance is a powerful tool for comparing probability distributions and is widely used for document classification and retrieval tasks in NLP. In particular, it is known as the word mover's distance (WMD) in the NLP community. WMD exhibits excellent performance for various NLP tasks; however, one of its limitations is its computational cost and thus is not useful for large-scale distribution comparisons. In this study, we propose a simple and effective nearest neighbor search based on the Wasserstein distance. Specifically, we employ the L1 embedding method based on the tree-based Wasserstein approximation and subsequently used the nearest neighbor search to efficiently find the $k$-nearest neighbors. Through benchmark experiments, we demonstrate that the proposed approximation has comparable performance to the vanilla Wasserstein distance and can be computed three orders of magnitude faster than the vanilla Wasserstein distance.

## 1 Introduction

Information retrieval is the process of extracting relevant information from large collections of data. It has applications in search engines, recommendation systems, and content analyses. An information retrieval task can be formulated as a $k$-nearest neighbor search problem. One major challenge in $k$-nearest neighbor search is the scalability of document retrieval tasks, where data can range from millions to billions of documents.

The bag-of-words (BOW) model is a widely used approach for document representation in natural language processing and information retrieval. It represents a document as a vector of word frequencies, disregarding the order and context of the words. This approach is simple and efficient; however, it suffers when two documents do not share common words. This is because the BOW model does not capture the semantic relationships between words and their context in the document, leading to poor retrieval performance when dealing with documents that are dissimilar in content.

Word mover's distance (WMD) (Kusner et al., 2015) and its variants (Yokoi et al., 2020; Huang et al., 2016) are document dissimilarity measures that can handle overlapping problems arising in the BOW model. WMD calculates the distance between two documents by measuring the minimum cost of transforming the word embeddings of one document into those of another (Kusner et al., 2015; Sato et al., 2022). However, WMD and its variants have a computational complexity of the cubic order, making them slow and impractical for large-scale information retrieval tasks involving millions or billions of documents.

To decrease computational complexity, a tree-based approximation of WMD known as tree-Wasserstein distance (TWD) can be used. QuadTree (Indyk and Thaper, 2003) and ClusterTree (Le et al., 2019) are techniques that enable tree construction for TWD and can be computed in linear time. Recently, Backurs et al. (2020) applied QuadTree to large-scale nearest neighbor searches. However, their performance is limited. Moreover, Backurs et al. (2020) proposed FlowTree, an algorithm that has improved performance but increased computation time. Additionally, existing approaches are based on pairwise comparisons and cannot directly compute one-to-many distributions. Hence, there is no optimal transport (OT) method that can be used for practical large-scale nearest neighbor search with high accuracy.

In this study, we introduce an OT-based nearest neighbor search for large-scale problems. Specifically, we combine three methods: the L1-regularized technique for learning the weights of the edges in a tree (Yamada et al., 2022), the sliced variant of TWD (Le et al., 2019), and the nearest neighbor search based on FAISS (Johnson et al., 2019). This combination enable the construction of a computationally efficient OT-based nearest neigh-

bor search that can be scaled to a large dataset. Through benchmark experiments, we demonstrated that the proposed framework can compute the nearest neighbors three orders of magnitude faster than the vanilla Wasserstein distance.

**Contribution:** The contribution of this paper is summarized below:

- A large-scale nearest neighbor search method based on OT is proposed.

- A simple yet memory-efficient representation of L1 embedding is presented.

- The proposed algorithm is demonstrated to be several orders of magnitude faster than WMD.

## 2 Proposed Framework

In this section, we propose the nearest neighbor search method based on TWD (Indyk and Thaper, 2003; Le et al., 2019).

### 2.1 Nearest Neighbor Search with Tree Wasserstein Distance

The TWD between the two discrete measures $\mu = \sum_{i=1}^{N_{\text{leaf}}} a_i \delta_{\boldsymbol{x}_i}$ and $\nu = \sum_{j=1}^{N_{\text{leaf}}} b_j \delta_{\boldsymbol{x}_j}$, with $\boldsymbol{a}^\top \mathbf{1} = 1$ and $\boldsymbol{b}^\top \mathbf{1} = 1$, can be expressed by L1 embedding (Takezawa et al., 2021):

$$W_{\mathcal{T}}(\mu, \nu) = \|\boldsymbol{u_a} - \boldsymbol{u_b}\|_1,$$

where $\boldsymbol{u_a} = \text{diag}(\boldsymbol{w})\boldsymbol{Ba}$ and $\boldsymbol{u_b} = \text{diag}(\boldsymbol{w})\boldsymbol{Bb}$. $\boldsymbol{w} \in \mathbb{R}_+^N$ is the edge weight of a tree, $\text{diag}(\boldsymbol{w}) \in \mathbb{R}_+^{N \times N}$ is the diagonal matrix whose diagonal elements are $\boldsymbol{w}$, and $\boldsymbol{B} \in \{0,1\}^{N \times N_{\text{leaf}}}$ is a tree parameter. If the node $i$ is the ancestor node of the leaf node $j$, $[\boldsymbol{B}]_{i,j} = 1$; othwerise, $[\boldsymbol{B}]_{i,j} = 0$. In addition, $N$ is the total number of nodes of a tree and $N_{\text{leaf}}$ is the number of leaf nodes.

Using TWD, the nearest neighbor search from $\{\mu_k\}_{k=1}^n$ with $\mu_k = \sum_i a_i^{(k)} \delta_{\boldsymbol{x}_i}$ can be written as

$$\widehat{k} = \underset{k \in \{1,2,\dots,n\}}{\text{argmin}} \|\boldsymbol{u_a} - \boldsymbol{u_{a_k}}\|_1.$$

Hence, the nearest neighbor search using TWD can be formulated as the nearest neighbor search with a L1 distance. In this study, we proposed a combination of the L1 embedding and the nearest neighbor methods. Specifically, for an efficient nearest neighbor search, we used the IndexFlat and GPUIndexFlat functions in the FAISS package.

L1 embedding for TWD is a well-known result in theoretical computer science (Indyk and Thaper,

2003). However, its performance is limited because of poor approximation of the original Wasserstein distance.

### 2.2 Word Mover's Distance Approximation with Tree

In this section, we explain how to approximate WMD using a tree. Note that WMD is a Wasserstein distance that uses word vectors to compute the distance $d(\boldsymbol{x}, \boldsymbol{x}')$.

The difference between WMD and TWD is the distance metric. Specifically, WMD uses L2 distance $d(\boldsymbol{x}, \boldsymbol{x}') = \|\boldsymbol{x} - \boldsymbol{x}'\|_2$, whereas TWD uses a tree metric (Le et al., 2019; Yamada et al., 2022). Thus, if $d(\boldsymbol{x}, \boldsymbol{x}') = d_{\mathcal{T}}(\boldsymbol{x}, \boldsymbol{x}')$, we can approximate WMD with TWD (i.e., $W_1(\mu, \nu) = W_{\mathcal{T}}(\mu, \nu)$).

**Proposition 1** *(Yamada et al., 2022) We denote $\boldsymbol{B} \in \{0,1\}^{N \times N_{\text{leaf}}} = [\boldsymbol{b}_1, \boldsymbol{b}_2, \dots, \boldsymbol{b}_{N_{\text{leaf}}}]$ and $\boldsymbol{b}_i \in \{0,1\}^N$. The shortest path distance between leaves $i$ and $j$ can be represented as*

$$d_{\mathcal{T}}(\boldsymbol{x}_i, \boldsymbol{x}_j) = \boldsymbol{w}^\top(\boldsymbol{b}_i + \boldsymbol{b}_j - 2\boldsymbol{b}_i \circ \boldsymbol{b}_j). \quad (1)$$

We stated the edge-weight estimation problem using the closed-form expression in Proposition 1. Our approach assumed that the tree was constructed using tree construction algorithms, such as QuadTree (Indyk and Thaper, 2003) and ClusterTree (Le et al., 2019), and we held the tree structure constant (fixed $\boldsymbol{B}$). Subsequently, the weight estimation problem can be expressed as (Yamada et al., 2022)

$$\widehat{\boldsymbol{w}} := \underset{\boldsymbol{w} \in \mathbb{R}_+^N}{\text{argmin}} \sum_{(i,j) \in \Omega} (d(\boldsymbol{x}_i, \boldsymbol{x}_j) - \boldsymbol{w}^\top \boldsymbol{z}_{i,j})^2 + \lambda \|\boldsymbol{w}\|_1,$$

where $\boldsymbol{z}_{i,j} = \boldsymbol{b}_i + \boldsymbol{b}_j - 2\boldsymbol{b}_i \circ \boldsymbol{b}_j$, and $\Omega$ denotes a set of indices. In this study, we randomly subsampled the indices to reduce computational cost.

### 2.3 TWD with efficient NN

In this study, we aimed to propose a large-scale similarity search; however, a simple nearest neighbor search method cannot effectively scale to large datasets. Thus, we employed an off-the-shelf efficient nearest neighbor search method called FAISS[1]. Specifically, we proposed using the L1 version of the FAISS function, as TWD is defined as the L1 distance. However, the dimensionality

---

[1] https://github.com/facebookresearch/faiss

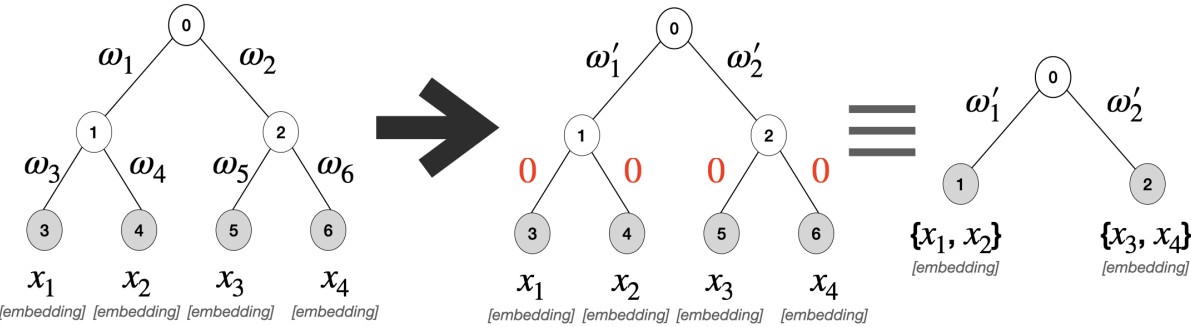

Figure 1: Illustration of a tree without leaf nodes.

of $z$ is substantially high because the dimensionality of the L1 embedding vector $z$ depends on the number of word vectors, which is large. Thus, directly using the tree embedding $z = \text{diag}(w)Ba$ is inefficient.

To address this issue, we proposed the use of the ClusterTree (Le et al., 2019) and disregard the leaf nodes (see Figure 1). This implies using only the internal nodes and is independent of the number of leaf nodes (i.e., the number of word vectors). For example, if we use ClusterTree with $K$ classes and depth $D$, the maximum number of internal nodes is $K + K^2 + \ldots + K^D$. As a rule of thumb, we can set $K \in \{3, 4, 5, 6\}$ and $D \in \{3, 4, 5, 6\}$.

Using only the internal nodes can significantly reduce the number of dimensions $z$. However, the representation power is significantly reduced. To address this issue, we proposed the use of multiple trees. Specifically, let $\bar{B}_t$ denote a ClusterTree parameter without leaf nodes; the sliced L1 embedding is expressed as

$$\bar{z} = \begin{pmatrix} \text{diag}(\bar{w}_1)\bar{B}_1 a \\ \text{diag}(\bar{w}_2)\bar{B}_2 a \\ \vdots \\ \text{diag}(\bar{w}_L)\bar{B}_L a \end{pmatrix},$$

where the weight for the $t$-th sliced weight $\bar{w}_t$ is trained as follows:

$$\bar{w}_t := \underset{w \in \mathbb{R}_+^{\bar{N}}}{\arg\min} \sum_{(i,j) \in \Omega} (d(x_i, x_j) - w^\top \bar{z}_{i,j}^{(t)})^2 + \lambda \|w\|_1,$$

and $\bar{z}_{i,j}^{(t)} = \bar{b}_i^{(t)} + \bar{b}_j^{(t)} - 2(\bar{b}_i^{(t)} \circ \bar{b}_j^{(t)}) \in \mathbb{R}^{\bar{N}_t}$ with $\bar{N}_t = K + K^2 + \ldots + K^D$. This modification can significantly reduce the training time.

## 3  Experiments

In this section, we assess the effectiveness of our proposed method using Twitter, BBCSport, Amazon, and Classic datasets.

### 3.1  Setup

We evaluated our proposed method for document classification tasks using the $k$-nearest neighbors (kNN) search. Our primary objective is to assess the accuracy and computation time of the tree-based method and compare them with those of the Wasserstein distance method. To calculate the Wasserstein distance, we used the Python optimal transport package[2] and considered the resulting accuracy and computation time as baselines. The bag-of-words (BOW) results are given for comparison.

We specifically evaluated the ClusterTree approach (Le et al., 2019) for the tree-based methods. The number of clusters and the maximum depth for all experiments were set to $K = 5$ and $D = 6$, respectively. Our initial step involved constructing a tree using an entire word embedding (word2vec) vector $X$, followed by the computation of the Wasserstein distance with the tree. We set the regularization parameter to $\lambda = 10^{-2}$ and explored different numbers of slices $T \in \{1, 3, 5, 10\}$. To solve the Lasso-based regression problem associated with the weight estimation problem, the SPAMS library[3] was used. In addition, the weight of the leaf nodes was set to zero for the leafless tree method (ll-TWD). For your reference, we conducted an additional evaluation using the QuadTree approach, and you can find the corresponding results in the appendix.

To compare the performance of all methods, we used the kNN search for varying values of $k \in \{1, 5, 10, 15\}$ and used GPU FAISS index with the L1 metric for the tree-based methods. By changing the random seed, the datasets were split five times into training and testing subsets such that 70% of each dataset was used for training. Twitter,

---

[2] https://pythonot.github.io/index.html
[3] http://thoth.inrialpes.fr/people/mairal/spams

| Methods | Twitter | | | BBCSport | | | Amazon | | | Classic | | |
|---|---|---|---|---|---|---|---|---|---|---|---|---|
| | 1 | 5 | 10 | 1 | 5 | 10 | 1 | 5 | 10 | 1 | 5 | 10 |
| WMD | 248.7 | 250.2 | 250.9 | 283.9 | 275.5 | 267.5 | 5382.8 | 5380.3 | 5380.5 | 5093.1 | 5079.4 | 5097.9 |
| BOW | 0.036 | 0.036 | 0.036 | 0.011 | 0.011 | 0.010 | 0.695 | 0.692 | 0.692 | 0.324 | 0.321 | 0.322 |
| Faiss TWD | 0.042 | 0.043 | 0.043 | 0.011 | 0.011 | 0.011 | 0.784 | 0.779 | 0.779 | 0.390 | 0.389 | 0.389 |
| Faiss ll-TWD n_slice=1 | 0.020 | 0.020 | 0.020 | 0.006 | 0.006 | 0.006 | 0.104 | 0.101 | 0.101 | 0.083 | 0.081 | 0.082 |
| Faiss ll-TWD n_slice=3 | 0.032 | 0.033 | 0.033 | 0.008 | 0.008 | 0.008 | 0.244 | 0.239 | 0.239 | 0.171 | 0.167 | 0.167 |
| Faiss ll-TWD n_slice=5 | 0.044 | 0.044 | 0.045 | 0.010 | 0.010 | 0.010 | 0.378 | 0.370 | 0.370 | 0.264 | 0.257 | 0.258 |
| Faiss ll-TWD n_slice=10 | 0.075 | 0.072 | 0.071 | 0.017 | 0.017 | 0.017 | 0.752 | 0.745 | 0.748 | 0.493 | 0.486 | 0.486 |

Table 1: Average testing time of the datasets used in the experiments for $k \in \{1, 5, 10\}$.

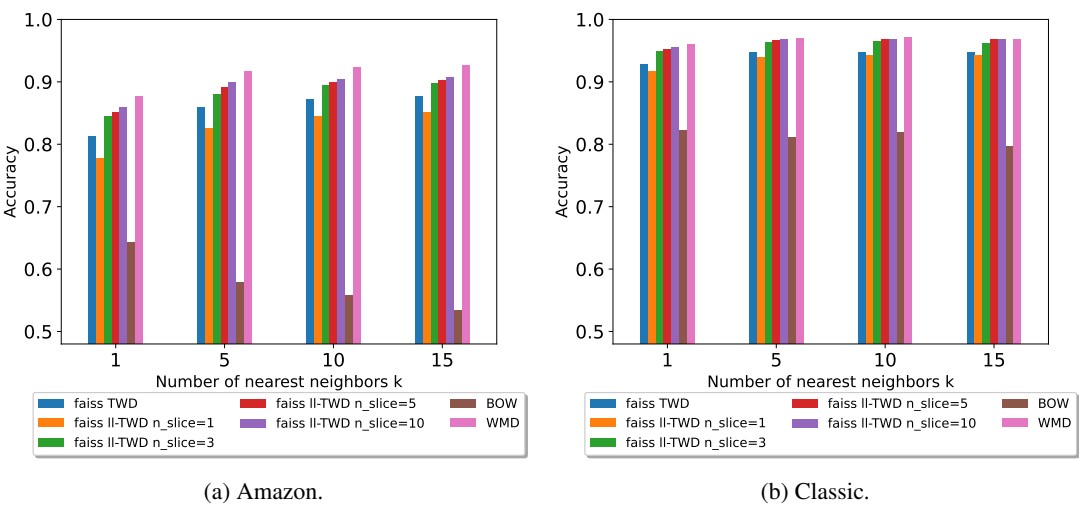

(a) Amazon.

(b) Classic.

Figure 2: Average accuracy of the datasets used in the experiments.

BBCSport, Amazon, and Classic[4] datasets used for the evaluation are presented in Table 2.

## 3.2 Results

The document classification experiments aimed to evaluate the effectiveness of tree-based methods in reducing the time required to classify data. To this end, we used four datasets and performed five different train-test splits on each dataset. For each split, we constructed a tree for the TWD-based methods and measured the accuracy and time required to classify the data from the testing set. The process of constructing a tree is a one-time task that enables the handling of all queries. The average time required to construct the tree using each method is shown in Figure 4.

The results of the experiments are presented in Table 1, Figure 2 and Figure 3. Table 1 shows the average time required to classify 30% of the data from each dataset for the $k$-nearest neighbors for $k \in \{1, 5, 10\}$, and Figure 2 illustrates the corresponding average accuracy. For information, we conducted an analysis of speed performance us-

ing the CPU FAISS index, and you can review the results in Table 3 located in the appendix. For $m \in \{1, 5\}$, Figure 3 gives the proportion of the $m$ first nearest neighbors of a given method that are actually the $m$ first nearest neighbors when using WMD. WMD is considered as a ground truth. As an illustration, within the Classic database, 46% of the data points from the testing set have the tree Wasserstein method with leaf nodes (faiss TWD method) sharing the same nearest neighbor as WMD.

For both Figures 2 and 3, the results presented are for Amazon and Classic datasets. A comprehensive overview of outcomes across all datasets is available in the appendix. The average time and accuracy and the top-$m$ score were calculated based on the five-time split for each dataset.

We observed that tree-based methods can reduce the classification time by a factor of $10^3$ compared to the WMD method. The fastest method is ll-TWD with $n\_slice = 1$, with the time required increasing with $n\_slice$. The faiss TWD method is faster than the leafless method when $n\_slice$ becomes large. In contrast, the accuracy improves

[4]https://github.com/gaohuang/S-WMD

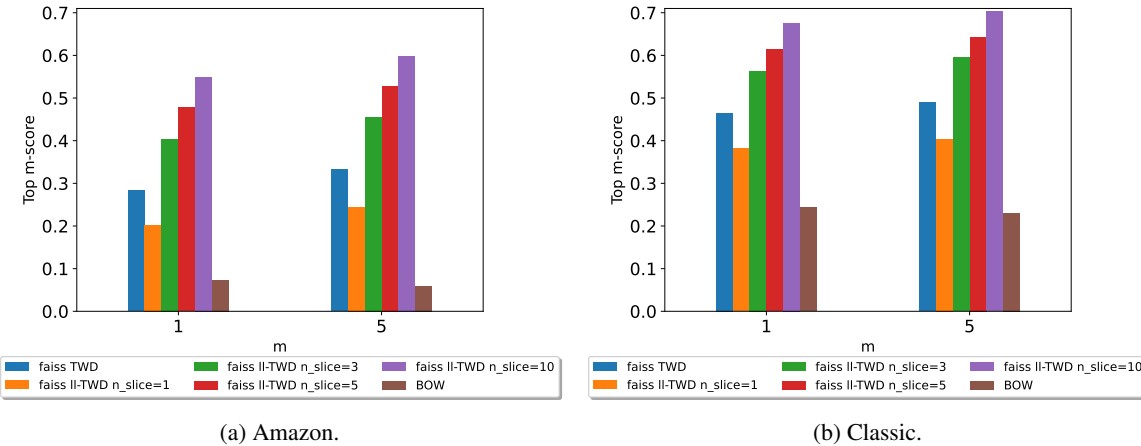

| (a) Amazon. | (b) Classic. |

Figure 3: Top-1 and Top-5 neighbors scores, with WMD as a ground truth, for the datasets used in the experiments.

for the leafless-tree methods when $n\_slice$ is large, as well as the top-$m$ score. For example, we observed that the fastest method among all the tested methods is ll-TWD with $n\_slice = 1$. However, it exhibits the lowest accuracy among all the methods presented in Figure 2, and the lowest top-$m$ score in Figure 3, BOW apart. As a result, a trade-off between test time and accuracy must be achieved.

In our experiments, the ll-TWD method with $n\_slice = 3$ balances between test time and accuracy. Consequently, it offers a suitable trade-off solution for classification tasks. Overall, our experiment demonstrates the effectiveness of tree-based methods in reducing the classification time while maintaining an acceptable level of accuracy.

## 4 Conclusion

In this study, we introduced a methodological improvement aimed at enhancing the computational speed of the Wasserstein distance approximation with trees. Specifically, we combined the L1-regularized technique to learn the weights of the edges in a tree (Yamada et al., 2022), the sliced-TWD (Le et al., 2019), and the efficient nearest neighbor search (Johnson et al., 2019). To further reduce the computational time, we disregarded the leaf nodes of the trees, as they can significantly reduce the dimensions of the L1 embedding vectors. The sliced TWD allowed us to deal with the decrease in representation power resulting from the leaf loss. Through benchmark experiments, we demonstrated that the proposed method can reduce computation time over $10^3$ times compared with the WMD method while maintaining a reasonable level of accuracy.

## Limitation

The proposed approach is based on the approximation of the Wasserstein distance; the performance is generally lower than that of the vanilla Wasserstein distance. Furthermore, while we have focused on improving WMD's computational efficiency, we acknowledge that there are contexts where WMD may not be the most suitable measure. Our aim in this work has been to offer a more efficient approach for computing WMD, especially in applications that prioritize vocabulary associations over syntactic structure, like content-based recommendation systems, plagiarism detection, and information retrieval. Eventually, the potential scalability of our method relies on both the scalability of the tree embedding technique and the chosen nearest neighbor package. One limitation in the majority of NN packages is a lack of support for sparse formats. Given that our tree embedding produces highly sparse vectors, addressing this limitation is crucial for handling larger datasets.

## Ethics/Broader impact

This is the first practical approach for nearest neighbor search using Wasserstein distance. Thus, by replacing a simple NN-based vector search with the proposed method, we can achieve a significant gain over simple vector-based NN methods. Thus, notwithstanding the simplicity of the approach, it can improve many NLP applications based on nearest neighbor search. There are no ethical issues associated with this study.

## Acknowledgement

Makoto Yamada was supported by MEXT KAK-ENHI Grant Number 20H04243.

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

## A  Related work

In this study, we considered an optimal transport-based nearest neighbor search for a large-scale dataset. Thus, we described the optimal transport (Wasserstein distance) and the nearest neighbor search (NN).

**Optimal transport and Wasserstein distance:** Optimal transport is widely used in machine learning, computer vision, and natural language processing tasks. The Monge formulation was originally proposed (Monge, 1781), in which it moves mass from one place to another. In NLP communities, the Kantorovich formulation is more widely used than the Monge formulation (Kantorovich, 1942; Peyré et al., 2019). In this study, we will focus on the Kantorovich formulations.

Kusner et al. (2015) proposed the word mover's distance (WMD) for NLP tasks. WMD is given as

$$W_1(\mu, \nu) = \min_{\mathbf{\Pi} \in U(\boldsymbol{a}, \boldsymbol{a}')} \sum_{i=1}^{m} \sum_{j=1}^{m'} \pi_{ij} d(\boldsymbol{x}_i, \boldsymbol{x}'_j),$$

where $\mu = \sum_i a_i \delta_{\boldsymbol{x}_i}$ and $\nu = \sum_j a'_j \delta_{\boldsymbol{x}'_j}$, $\boldsymbol{a}^\top \mathbf{1} = 1$ and $\boldsymbol{a}'^\top \mathbf{1} = 1$. In addition, $\boldsymbol{x} \in \mathbb{R}^d$ is a word vector, $\mathbf{\Pi}$ is a transport plan, and $U$ denotes a set of joint probability distributions.

WMD is highly used in various natural language processing tasks, including document classification (Kusner et al., 2015; Sato et al., 2022), text generation, and machine translation. However, the computation of WMD is cubic with respect to the number of data points. Therefore, it cannot effectively scale to large datasets.

To speed up the computation of WMD (or Wasserstein distance), we can use the Sinkhorn algorithm (Cuturi, 2013), which solves entropic regularized optimal transport problems. Owing to the Sinkhorn algorithm, WMD can be efficiently solved in quadratic time.

To further speed up the computation of Wasserstein distance, Wasserstein distance on a tree is useful (Indyk and Thaper, 2003). Specifically, it first embeds data points in a tree and computes the optimal transport on the tree. The advantage of tree-based optimal transport is that the Wasserstein distance can be obtained analytically and computed in linear cost. Le et al. (2019) proposed a tree sliced variants of Wasserstein distance and showed that the performance significantly be improved by using multiple trees. Moreover, TWD can be represented by the L1 distance between two embedded vectors

(Takezawa et al., 2021). Thus, it can be efficiently computed using GPUs. The tree-based method is computationally efficient; however, it does not approximate the Wasserstein distance using an arbitrary metric. To solve this issue, Yamada et al. (2022) recently proposed a regression-based approach to approximate Wasserstein distance using TWD.

**Nearest Neighbor Search:** Nearest neighbor search is a fundamental technique used in various fields, including natural language processing (NLP), machine learning, data mining, information retrieval, and computer vision. The nearest neighbor search is the key technique in document retrieval, which is a typical application of NLP. It is also used for image retrieval in computer vision applications.

One of the key challenges of the nearest neighbor search is the computational cost. Because the number of documents increases dramatically, a vanilla nearest neighbor search is not applicable to such large-scale data. To handle the scalability issue, an approximate nearest neighbor search, which efficiently finds the nearest neighbors by approximating the distance computation, is heavily used.

One standard approach is to use kd-tree to partition the space and quickly find nearest neighbors (Bentley, 1975). The other techniques are based on locality-sensitive hashing (Datar et al., 2004). Recently, FAISS has been proposed for finding nearest neighbors from billion-scale datasets using GPUs (Johnson et al., 2019). Although approximate nearest neighbor search algorithms are heuristic approaches, they can efficiently search for nearest neighbors. Recently, several useful ANN packages have been developed, including the FAISS[5], AN-NOY [6], and SCANN [7].

| | Number of documents $n$ | Number of unique words $N_{leaf}$ |
|---|---|---|
| Twitter | 3108 | 4489 |
| BBCsport | 737 | 10103 |
| Amazon | 8000 | 30249 |
| Classic | 7093 | 18080 |

Table 2: Number of documents and unique words of the datasets used for the experiments.

[5]https://github.com/facebookresearch/faiss
[6]https://github.com/spotify/annoy
[7]https://github.com/google-research/google-research/tree/master/scann

## B  Additional results

Figure 4 shows the tree construction time that is performed only once for each method. The smaller the $n\_slice$, the faster the construction time.

Table 3 gives the average time required to classify 30% of the data from each dataset for the $k$-nearest neighbors for $k \in \{1, 5, 10\}$ when using a CPU FAISS index. The smaller the $n\_slice$, the faster the construction time.

The average accuracy for all datasets is presented in Figure 5.

Figure 6 reports the results for all datasets of another performance indicator : top-$m$ neighbors score. Top-$m$ neighbors score, here for $m \in \{1, 5\}$, gives the proportion of the $m$ first nearest neighbors of a given method that are actually the $m$ first nearest neighbors when using WMD. WMD is considered as a ground truth.

Ultimately, Figures 7, 8 and Table 4 present the tree construction time, the average accuracy and the average time required to classify 30% of the data from each database generated using the QuadTree approach. These results underscore that the ClusterTree approach for the tree construction gives better performance.

**Algorithm 1** Sliced weight estimation with trees.

1: **Input:** The matrix $\boldsymbol{X}$, the regularization parameter $\lambda_1 \geq 0$, and a set of indices $\Omega$.
2: **for** $t = 1, 2, \ldots, T$ **do**
3:     random.seed($i$)
4:     $\boldsymbol{B}_t := \text{ClusterTree}(\boldsymbol{X})$
5:     $\bar{\boldsymbol{B}}_t := \text{RemoveLeaf}(\boldsymbol{B}_t)$
6:     Compute $\boldsymbol{z}_{i,j}^{(t)}$ from $\bar{\boldsymbol{B}}_t$ and $d(\boldsymbol{x}_i, \boldsymbol{x}_j), (i, j) \in \Omega$
7:     $\bar{\boldsymbol{w}}_t := \text{argmin}_{\boldsymbol{w} \in \mathbb{R}_+^{\bar{N}_t}} \sum_{(i,j) \in \Omega} (d(\boldsymbol{x}_i, \boldsymbol{x}_j) - \boldsymbol{w}^\top \bar{\boldsymbol{z}}_{i,j}^{(t)})^2 + \lambda \|\boldsymbol{w}\|_1.$
8: **end for**
9: **return** $\{(\bar{\boldsymbol{B}}_t, \bar{\boldsymbol{w}}_t)\}_{t=1}^T$

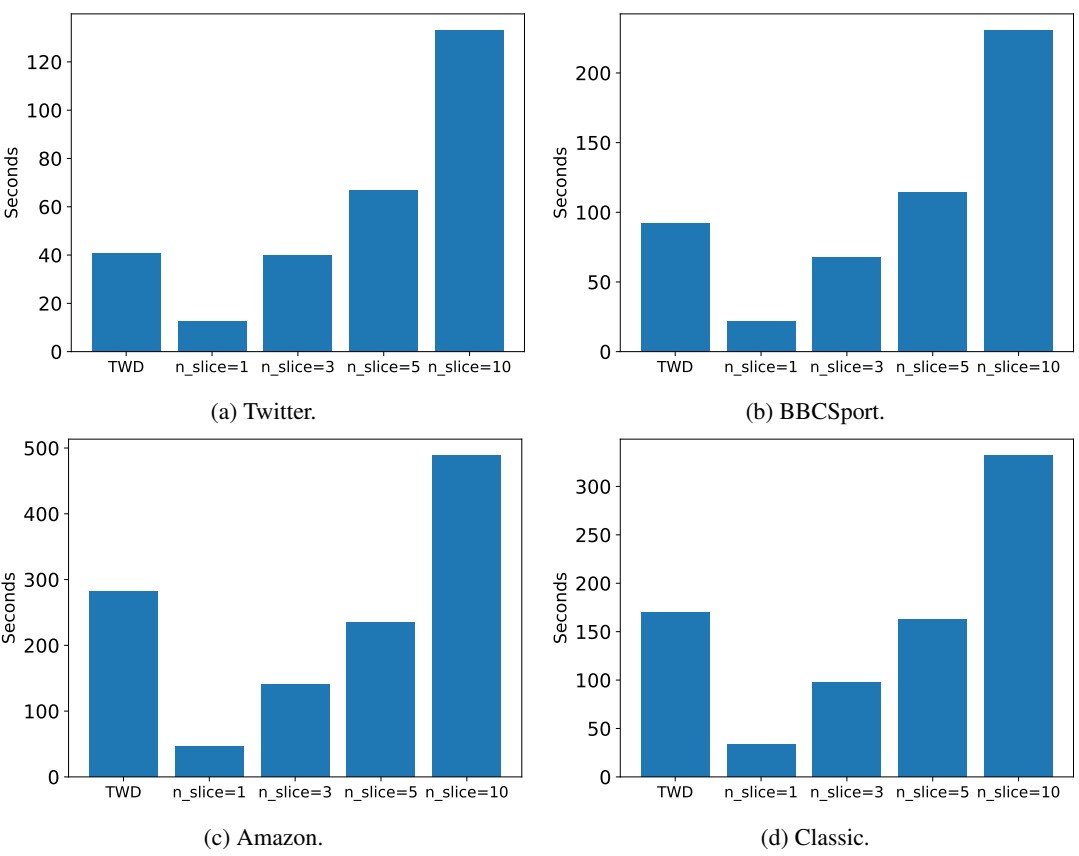

(a) Twitter.          (b) BBCSport.

(c) Amazon.          (d) Classic.

Figure 4: One-time tree construction time (expressed in seconds) of the datasets used in the experiments.

| Methods | Twitter | | | BBCSport | | | Amazon | | | Classic | | |
|---|---|---|---|---|---|---|---|---|---|---|---|---|
| | 1 | 5 | 10 | 1 | 5 | 10 | 1 | 5 | 10 | 1 | 5 | 10 |
| Faiss TWD | 3.969 | 3.917 | 4.033 | 0.389 | 0.347 | 0.351 | 174.105 | 169.923 | 165.540 | 81.241 | 80.908 | 81.603 |
| Faiss ll-TWD n_slice=1 | 0.430 | 0.428 | 0.449 | 0.044 | 0.044 | 0.045 | 14.749 | 15.265 | 15.228 | 10.330 | 9.860 | 9.997 |
| Faiss ll-TWD n_slice=3 | 2.224 | 2.319 | 2.336 | 0.156 | 0.159 | 0.139 | 47.057 | 47.426 | 47.380 | 30.582 | 30.345 | 30.283 |
| Faiss ll-TWD n_slice=5 | 4.205 | 4.331 | 4.298 | 0.325 | 0.307 | 0.308 | 78.731 | 80.303 | 80.349 | 51.749 | 51.749 | 51.580 |
| Faiss ll-TWD n_slice=10 | 9.171 | 8.849 | 9.135 | 0.790 | 0.778 | 0.809 | 163.348 | 160.211 | 159.874 | 102.815 | 102.325 | 102.379 |

Table 3: Average testing time of the datasets used in the experiments for $k \in \{1, 5, 10\}$ using a CPU faiss index.

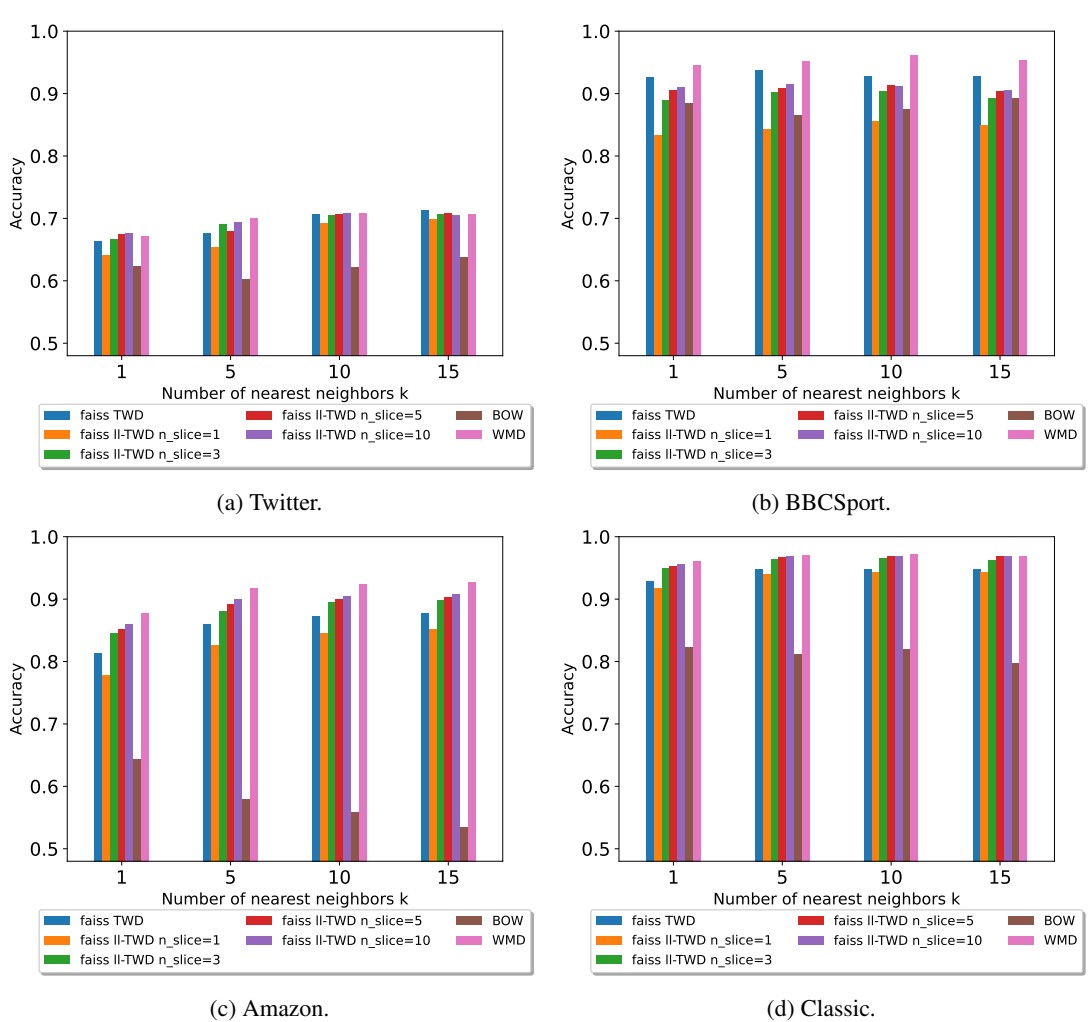

(a) Twitter.

(b) BBCSport.

(c) Amazon.

(d) Classic.

Figure 5: Average accuracy of the datasets used in the experiments.

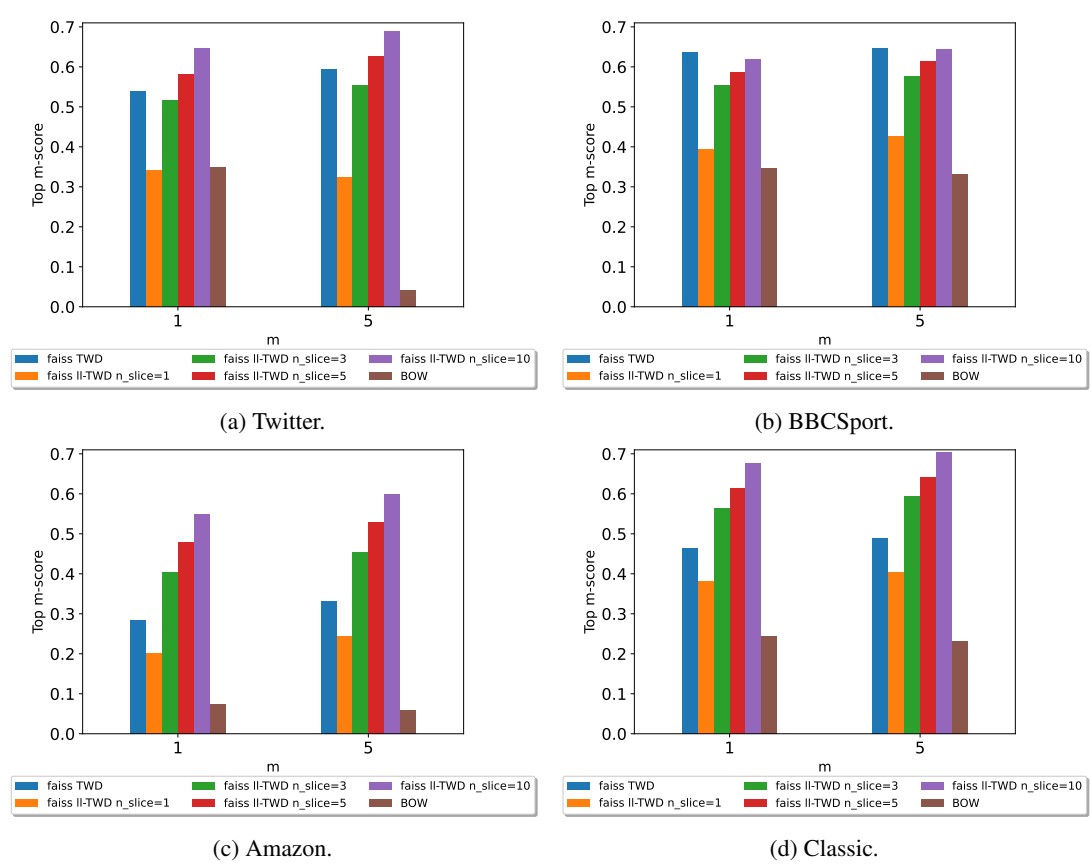

(a) Twitter.

(b) BBCSport.

(c) Amazon.

(d) Classic.

Figure 6: Top-1 and Top-5 neighbors scores, with WMD as a ground truth, for the datasets used in the experiments.

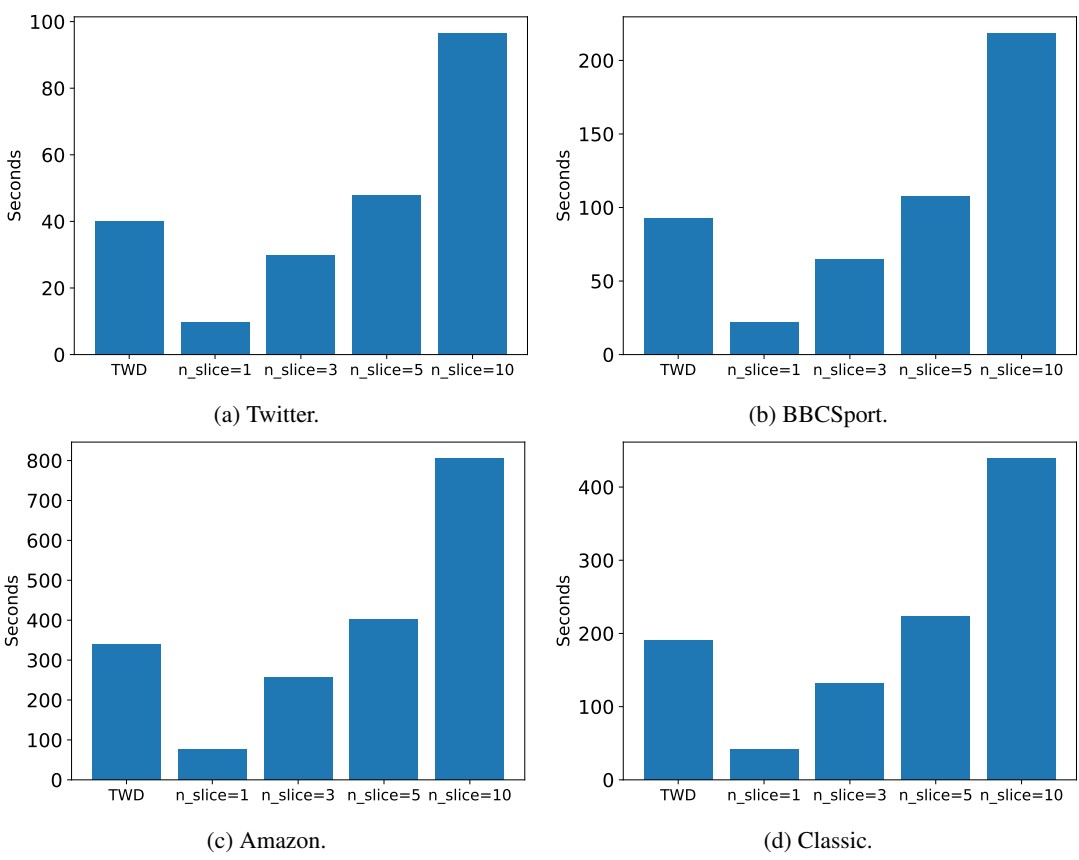

(a) Twitter.

(b) BBCSport.

(c) Amazon.

(d) Classic.

Figure 7: One-time tree construction time (expressed in seconds) of the datasets used in the experiments for the QuadTree approach.

| Methods | Twitter | | | BBCSport | | | Amazon | | | Classic | | |
|---|---|---|---|---|---|---|---|---|---|---|---|---|
| | 1 | 5 | 10 | 1 | 5 | 10 | 1 | 5 | 10 | 1 | 5 | 10 |
| WMD | 248.7 | 250.2 | 250.9 | 283.9 | 275.5 | 267.5 | 5382.8 | 5380.3 | 5380.5 | 5093.1 | 5079.4 | 5097.9 |
| BOW | 0.036 | 0.036 | 0.036 | 0.011 | 0.011 | 0.010 | 0.695 | 0.692 | 0.692 | 0.324 | 0.321 | 0.322 |
| Faiss TWD | 0.045 | 0.051 | 0.051 | 0.012 | 0.012 | 0.012 | 0.799 | 0.803 | 0.799 | 0.642 | 0.647 | 0.637 |
| Faiss ll-TWD n_slice=1 | 0.020 | 0.021 | 0.022 | 0.008 | 0.009 | 0.008 | 0.117 | 0.115 | 0.117 | 0.096 | 0.090 | 0.095 |
| Faiss ll-TWD n_slice=3 | 0.030 | 0.031 | 0.030 | 0.012 | 0.009 | 0.009 | 0.298 | 0.300 | 0.302 | 0.283 | 0.266 | 0.248 |
| Faiss ll-TWD n_slice=5 | 0.036 | 0.035 | 0.043 | 0.009 | 0.010 | 0.009 | 0.489 | 0.484 | 0.486 | 0.407 | 0.404 | 0.403 |
| Faiss ll-TWD n_slice=10 | 0.056 | 0.058 | 0.068 | 0.018 | 0.021 | 0.016 | 0.925 | 0.918 | 0.923 | 0.736 | 0.764 | 0.756 |

Table 4: Average testing time of the datasets used in the experiments for $k \in \{1, 5, 10\}$ for the QuadTree approach.

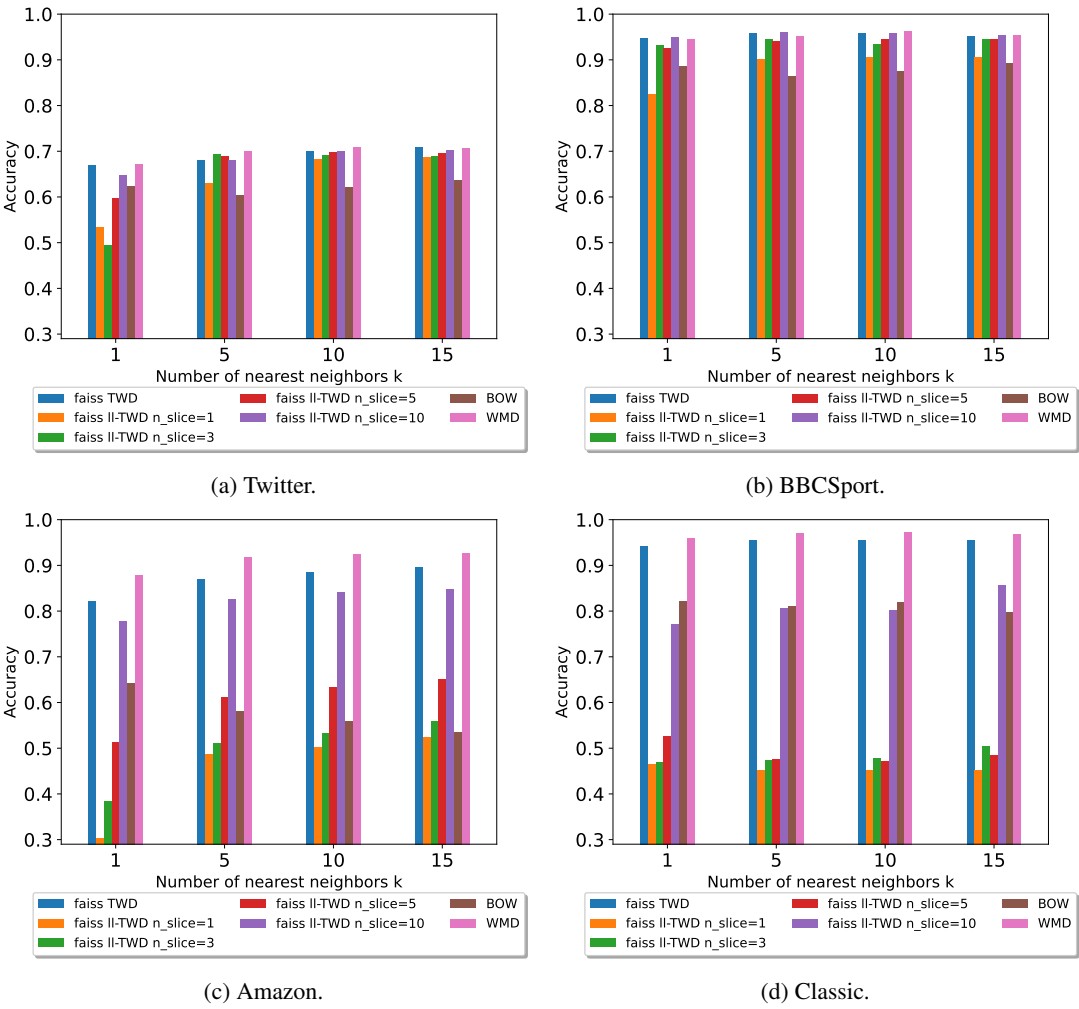

(a) Twitter.             (b) BBCSport.

(c) Amazon.             (d) Classic.

Figure 8: Average accuracy of the datasets used in the experiments for the QuadTree approach.