# OpenReview forum: "Large-scale similarity search with Optimal Transport"
_EMNLP/2023/Conference — EMNLP 2023 Main_

### Official Review · Reviewer_8vBA · 2023-07-31

**Soundness:** 3

**Excitement:**

4: Strong: This paper deepens the understanding of some phenomenon or lowers the barriers to an existing research direction.

**Paper Topic And Main Contributions:**

This paper proposes a new method to speed up the computation of the word mover's distance. The method is based on the L1 embedding method and the tree-based Wasserstein approximation. Based on the kNN classification results on four datasets, the proposed method is comparable to the vanilla Wasserstein distance in terms of performance and is much faster in inference speed.

**Questions For The Authors:**

A. The paper mentions using "IndexFlat" for approximated nearest neighbor search, this index actually performs exhaustive search and would not have any speedup.

After rebuttal: thanks the authors for the clarification on this point.

**Reasons To Accept:**

1. The motivation of the paper is clear. How to efficiently compute WMD is an important problem in many areas.
2. The proposed method is backed by detailed mathematical derivations and analysis.
3. Experiments on four datasets show some advantages of inference speed.

**Reasons To Reject:**

1. The four datasets used for evaluation are small. The largest one only has 8k documents. It is not clear whether the proposed method can scale to larger datasets.
2. Word mover's distance is not a very accurate measure of document similarity. It ignores the word order and the syntactic structure of the documents. Thus, improving computation of WMD efficiency may not be very useful in many NLP applications.

**Reproducibility:**

3: Could reproduce the results with some difficulty. The settings of parameters are underspecified or subjectively determined; the training/evaluation data are not widely available.

**Reviewer Confidence:**

3: Pretty sure, but there's a chance I missed something. Although I have a good feel for this area in general, I did not carefully check the paper's details, e.g., the math, experimental design, or novelty.

---

> ### Author Rebuttal · Authors · 2023-08-29
>
> > The four datasets used for evaluation are small. The largest one only has 8k documents. It is not clear whether the proposed method can scale to larger datasets.
>
> Thank you for your response. We were able to assess the performance of our approach using small to medium-sized datasets. Our method involves combining tree embedding with readily available nearest neighbor search packages. The potential scalability of our method relies on both the scalability of the tree embedding technique and the chosen nearest neighbor package.
>
> Regarding tree embedding, our current implementation effectively scales up to handling around $10^4$ documents. However, extending this scalability to accommodate millions to billions of documents poses challenges. We are actively exploring the incorporation of efficient clustering algorithms to achieve this scalability goal. Nevertheless, refining the weight estimation process for the tree methods remains an ongoing task.
>
> Additionally, we've encountered a limitation in the majority of nearest neighbor packages, as they lack support for sparse formats. Given that our tree embedding produces highly sparse vectors, addressing this limitation is crucial for handling larger datasets. It's worth noting that devising an efficient method for approximated nearest neighbor search with sparse data is a vibrant area of ongoing research.
>
> We will include these discussion in the final version.
>
> > Word mover's distance is not a very accurate measure of document similarity. It ignores the word order and the syntactic structure of the documents. Thus, improving computation of WMD efficiency may not be very useful in many NLP applications.
>
> Thank you for your comments. Introducing the order information in tree Wasserstein distance is indeed an interesting problem. However, compared to the standard Wasserstein distance formulation, it is difficult to incorporate the positional information. It needs more research to overcome this problem. We will add more discussion in the final version.
>
> Moreover, while our focus has been on improving the computational efficiency of WMD, we acknowledge that there are contexts where WMD may not be the most suitable measure due to the specific limitations you highlighted. Our intention in this work has been to offer a more efficient computation approach for WMD that could still benefit certain applications, especially those where capturing vocabulary associations is of greater importance than maintaining syntactic structure such as in content-based recommendation systems, plagiarism detection, and information retrieval. However, we will ensure to clarify these aspects and discuss the limitations of WMD in more detail in our final version.
>
> > A. The paper mentions using "IndexFlat" for approximated nearest neighbor search, this index actually performs exhaustive search and would not have any speedup.
>
> Thank you for your comment. Faiss library allows approximated nearest neighbor search but "IndexFlat" does perform efficient exhaustive search. This will be modified in the paper. At the moment "IndexFlat" is the only index capable of accommodating the essential L1 metric needed for TWD. While it does perform exhaustive search, a data structure in RAM is built which performs efficiently the argmin search on the index and reduce the computation time. This computation time could further be reduced with the adaptation of other indexes to the L1 metric.

---

### Official Review · Reviewer_Dg6r · 2023-08-04

**Typos Grammar Style And Presentation Improvements:** In line 105, B_{I,j} should be B_{i,j}.
**Soundness:** 4

**Excitement:**

3: Ambivalent: It has merits (e.g., it reports state-of-the-art results, the idea is nice), but there are key weaknesses (e.g., it describes incremental work), and it can significantly benefit from another round of revision. However, I won't object to accepting it if my co-reviewers champion it.

**Missing References:**

[1] Wang, Z., Zhou, D., Yang, M., Zhang, Y., Rao, C., & Wu, H. (2020, September). Robust document distance with Wasserstein-fisher-rao metric. In Asian Conference on Machine Learning (pp. 721-736). PMLR.

[2] Sato, R., Yamada, M., & Kashima, H. (2020). Fast unbalanced optimal transport on a tree. Advances in neural information processing systems, 33, 19039-19051.

**Paper Topic And Main Contributions:**

In this paper, the authors proposed to combine the fast estimation of L1 neighboring distance with Tree Wasserstein Distance. It is also natural to use libraries like FAISS by viewing TWD is essential L1 metric. Technically, it is a practical skill to accelerate the linear approximation of the computation.

**Questions For The Authors:**

1. Can FAISS-based tree Wasserstein acceleration be applied to unbalanced optimal transport document distance [1] or some already known tree approximation [2]?
2. Can we train weights $w$ on one dataset and transfer them to the other dataset?

[1] Wang, Z., Zhou, D., Yang, M., Zhang, Y., Rao, C., & Wu, H. (2020, September). Robust document distance with Wasserstein-fisher-rao metric. In Asian Conference on Machine Learning (pp. 721-736). PMLR.

[2] Sato, R., Yamada, M., & Kashima, H. (2020). Fast unbalanced optimal transport on a tree. Advances in neural information processing systems, 33, 19039-19051.

**Reasons To Accept:**

1. The improvement of efficiency brought by FAISS is significant, while the performance drop compared to original TWD with cluster tree is somewhat not very large.

**Reasons To Reject:**

Despite its technical soundness, the potential impact of this work is limited from two aspects:

1. The cluster-tree TWD and other TWD variants. This paper only combines the FAISS and the cluster-tree TWD. There are no comparisons with other strong TWD baselines such as directly using word embeddings in the quadtree data structure.

2. Moreover, the potential impact of this work could be limited if the optimal transport-based search doesn't outperform the sentence embedding-based search. Approximate OT distances in L1 space might be useless under the hypothesis where some sort of sentence embeddings (possibly generated by LMs) can outperform OT distances, and almost surely outperform approximate OT. Though this hypothesis is not now fully validated, the current NLP community seems to prefer the LM-based approach.

**Reproducibility:**

3: Could reproduce the results with some difficulty. The settings of parameters are underspecified or subjectively determined; the training/evaluation data are not widely available.

**Reviewer Confidence:**

4: Quite sure. I tried to check the important points carefully. It's unlikely, though conceivable, that I missed something that should affect my ratings.

---

> ### Author Rebuttal · Authors · 2023-08-29
>
> > The cluster-tree TWD and other TWD variants. This paper only combines the FAISS and the cluster-tree TWD. There are no comparisons with other strong TWD baselines such as directly using word embeddings in the quadtree data structure.
>
> We quickly run the Quadtree experiments. The results with Quadtree are presented in the following Tables 1, 2 and 3. Compared to the Clustertree, we found that the performances of Quadtree are lower.
>
> Table 1 : Average accuracy of the datasets used in the experiments for k $\in$ \{1,5,10\} with quadtree.
> |         |       | Twitter |       |      | BBCSport |       |      | Amazon |       |      | Classic |       |
> |:---:|:---:|:---:|:---:|:---:|:---:|:---:|:---:|:---:|:---:|:---:|:---:|:---:
> | Methods | 1     | 5       | 10    | 1     | 5        | 10    | 1     | 5      | 10    | 1     | 5       | 10    |
> |Faiss TWD | 0.669 |0.680 |0.701 |0.948 |0.958 |0.959 |0.822 |0.869 |0.886 |0.941 |0.954 | 0.956 |
> |Faiss ll-TWD $n\_{slice}$=1 | 0.533 |0.629 |0.683 |0.824 |0.901 |0.905 |0.303 |0.487 |0.503 |0.464 |0.451 |  0.451 |
> |Faiss ll-TWD $n\_{slice}$=3 |0.495 |0.693 |0.692 |0.931 |0.944 |0.934 |0.383 |0.511 |0.533 |0.470 |0.474 |  0.477|
> |Faiss ll-TWD $n\_{slice}$=5 | 0.597 |0.689 |0.697 |0.925 |0.941 |0.944 |0.514 |0.612 |0.634 |0.526 |0.475 |  0.472 |
> |Faiss ll-TWD $n\_{slice}$=10 |0.648 |0.681 |0.701 |0.950 |0.960 |0.958 |0.778 |0.826 |0.842 |0.772 |0.807 |  0.802|
>
>
> Table 2 : Average testing time of the datasets used in the experiments for k $\in$ \{1,5,10\} with quadtree.
> |         |       | Twitter |       |      | BBCSport |       |      | Amazon |       |      | Classic |       |
> |:---:|:---:|:---:|:---:|:---:|:---:|:---:|:---:|:---:|:---:|:---:|:---:|:---:
> | Methods | 1     | 5       | 10    | 1     | 5        | 10    | 1     | 5      | 10    | 1     | 5       | 10    |
> | Faiss TWD |   0.045 |   0.051 |   0.051 |   0.012 |   0.012 |   0.012 |   0.799 |   0.803 |   0.799 |   0.642 |   0.647 |   0.637 |
> | Faiss ll-TWD $n_{slice}$=1 |   0.020 |   0.021 |   0.022 |   0.008 |   0.009 |   0.008 |   0.117 |   0.115 |   0.117 |   0.096 |   0.090 |   0.095 |
> | Faiss ll-TWD $n_{slice}$=3 |   0.030 |   0.031 |   0.030 |   0.012 |   0.009 |   0.009 |   0.298 |   0.300 |   0.302 |   0.283 |   0.266 |   0.248 |
> | Faiss ll-TWD $n_{slice}$=5 |   0.036 |   0.035 |   0.043 |   0.009 |   0.010 |   0.009 |   0.489 |   0.484 |   0.486 |   0.407 |   0.404 |   0.403 |
> | Faiss ll-TWD $n_{slice}$=10 |   0.056 |   0.058 |   0.068 |   0.018 |   0.021 |   0.016 |   0.925 |   0.918 |   0.923 |   0.736 |   0.764 |   0.756 |
>
> Table 3 : One-time tree construction time (expressed in seconds) with quadtree for the datasets used in the experiments.
> |         | Twitter | BBCSport | Amazon | Classic |
> |:---:|:---:|:---:|:---:|:---:
> | Faiss TWD                   |  40.2    |  93.1     |  339.1  |  190.1  |
> | Faiss ll-TWD $n_{slice}$=1  |  9.7     |  22.2     |  76.3   |  41.6    |
> | Faiss ll-TWD $n_{slice}$=3  |  30.0    |  64.9     |  256.7 |  132.4  |
> | Faiss ll-TWD $n_{slice}$=5  |  48.0    |  107.5    |  402.9  |  222.8  |
> | Faiss ll-TWD $n_{slice}$=10 |  96.6    |  218.7    |  806.0  |  439.5 |
>
>
> > Moreover, the potential impact of this work could be limited if the optimal transport-based search doesn't outperform the sentence embedding-based search. Approximate OT distances in L1 space might be useless under the hypothesis where some sort of sentence embeddings (possibly generated by LMs) can outperform OT distances, and almost surely outperform approximate OT. Though this hypothesis is not now fully validated, the current NLP community seems to prefer the LM-based approach.
>
> Thank you for acknowledging the significance of the problem. Indeed, we consider the Optimal Transport (OT) based distance to be a potent metric, and it would be intriguing to ascertain its validity within the context of the existing literature on Language Models (LMs). While our current paper is focused on presenting a computationally efficient nearest neighbor search rooted in OT, we acknowledge that the broader goal of validating this approach within the LM literature remains a future direction. We intend to delve deeper into this aspect and augment our discussion in the final version.
>
> We wish to underscore that the Wasserstein distance itself is a foundational technique with diverse applications in Natural Language Processing (NLP) tasks. Hence, despite the superior performance of sentence embeddings over the OT based method, it remains pertinent to cultivate a computationally efficient approach.
>
> > Can FAISS-based tree Wasserstein acceleration be applied to unbalanced optimal transport document distance [1] or some already known tree approximation [2]?
>
> To compute the unbalanced OT, we need to solve optimization and cannot get analytical solution. Thus, it is not sure whether we can directly apply UOT for vector search. This is a very interesting future work.
>
> > Can we train weights  on one dataset and transfer them to the other dataset?
>
> It is possible to train weights on one dataset and transfer them to the other dataset as long as the vocabulary of the new dataset is included in the dataset used to train weights.

---

### Official Review · Reviewer_ZksN · 2023-08-05

**Typos Grammar Style And Presentation Improvements:** Maybe worth using L1 everywhere to be…
**Soundness:** 4

**Excitement:**

4: Strong: This paper deepens the understanding of some phenomenon or lowers the barriers to an existing research direction.

**Missing References:**

n/a

**Paper Topic And Main Contributions:**

This paper is about word mover's distance (WMD), a technique to compare two documents of varying length using their word embeddings. WMD can be used for kNN-based methods, but it is slow. The authors present an approximate WMD using faster nearest neighbor lookup (through FAISS), achieving a 1000x speed up and while retaining most of the accuracy of the exact method.

**Questions For The Authors:**

What is speed performance when using a CPU faiss index? The GPU index may become untenable with larger data.

Have you considered a continuous bag of words baseline? For example by averaging the word vectors then performing kNN? Kusner et al shows this does not work well, but it could still be interesting to sanity check and/or include as a row in the table to strengthen the motivation for using WMD.

**Reasons To Accept:**

1. WMD is useful for comparing documents but perhaps impractical, which is why it is common to resort to stuff documents into a single embedding. As NLP applications get more ambitious, documents are longer and perhaps it is better to use multiple embeddings. This paper can enable efficient use of WMD for applications like this.

2. The results are encouraging for both accuracy and speed on multiple datasets. Although I'm not sure why

3. The paper is also informative on the tradeoffs between accuracy and speed, which should be helpful for practitioners.

**Reasons To Reject:**

1. It would have been nice to include one or more non-WMD baselines. That being said, many such baselines are included in Kusner et al in favor of WMD (although embeddings have likely improved since then).

2. The datasets explored are still fairly small. Real world datasets can easily have thousands or millions of documents.

3. (low priority) It would be nice to see performance of WMD on additional tasks besides knn classification. For example, to retrieve documents for the BEIR tasks (perhaps using the approach from Hyde) or when retrieving passages in open domain QA.

BEIR https://arxiv.org/abs/2104.08663

Hyde https://arxiv.org/abs/2212.10496

Open-domain QA w/ retrieval https://arxiv.org/abs/2301.12652

**Reproducibility:**

5: Could easily reproduce the results.

**Reviewer Confidence:**

4: Quite sure. I tried to check the important points carefully. It's unlikely, though conceivable, that I missed something that should affect my ratings.

---

> ### Author Rebuttal · Authors · 2023-08-29
>
> > It would have been nice to include one or more non-WMD baselines. That being said, many such baselines are included in Kusner et al in favor of WMD (although embeddings have likely improved since then).
>
> The results with BOW are presented in Tables 1 and 2. As you can see, the proposed method outperforms BOW. Note that the BOW is computed using CPU; the computation speed is slower than that of the proposed TWD. If we use GPUs for BOW, we can surely get almost similar computational speed of the proposed method.
>
> Table 1 : BOW average accuracy of the datasets used in the experiments for k $\in$ \{1,5,10\}.
> |         |       | Twitter|       |      | BBCSport |       |      | Amazon |       |      | Classic |       |
> |:---:|:---:|:---:|:---:|:---:|:---:|:---:|:---:|:---:|:---:|:---:|:---:|:---:|
> | Methods | 1     | 5       | 10    | 1     | 5        | 10    | 1     | 5      | 10    | 1     | 5       | 10    |
> | BOW     | 0.624 | 0.603   | 0.622 | 0.885 | 0.865    | 0.875 | 0.643 | 0.580  | 0.559 | 0.822 | 0.811   | 0.820 |
>
> Table 2 : Average testing time of the datasets used in the experiments for k $\in$ \{1,5,10\}.
> |         |       | Twitter |       |      | BBCSport |       |      | Amazon |       |      | Classic |       |
> |:---:|:---:|:---:|:---:|:---:|:---:|:---:|:---:|:---:|:---:|:---:|:---:|:---:
> | Methods | 1     | 5       | 10    | 1     | 5        | 10    | 1     | 5      | 10    | 1     | 5       | 10    |
> | WMD | 248.7 |   250.2 |   250.9 |   283.9 |   275.5 |   267.5 |   5382.8 |   5380.3 |   5380.5 |  5093.1 |   5079.4 |   5097.9 |
> | BOW |   0.658 |   0.646 |   0.647 |   0.105 |   0.111 |   0.109 |   27.623 |   27.653 |   27.488 | 12.943 |   12.595 |   12.111 |
> |  Faiss TWD |   0.042 |   0.043 |   0.043 |   0.011 |   0.011 |   0.011 |   0.784 |   0.779 |   0.779 |  0.390 |   0.389 |   0.389 |
> | Faiss ll-TWD $n\_{slice}=1$ | 0.020 |   0.020 |   0.020 |   0.006 |   0.006 |   0.006 |   0.104 |   0.101 |  0.101 |   0.083 |   0.081 |   0.082 |
> | Faiss ll-TWD $n\_{slice}=3 $ |   0.032 |   0.033 |   0.033 |   0.008 |   0.008 |   0.008 |   0.244 |   0.239 | 0.239 |   0.171 |   0.167 |   0.167 |
> | Faiss ll-TWD $n\_{slice}=5$ |   0.044 |   0.044 |   0.045 |   0.010 |   0.010 |   0.010 |   0.378 |   0.370 | 0.370 |   0.264 |   0.257 |   0.258 |
> | Faiss ll-TWD $n\_{slice}=10$ |   0.075 |   0.072 |   0.071 |   0.017 |   0.017 |   0.017 |   0.752 |   0.745 |   0.748 |   0.493 |   0.486 |   0.486 |
>
> > The datasets explored are still fairly small. Real world datasets can easily have thousands or millions of documents.
>
>
> Thank you for your response. We were able to assess the performance of our approach using small to medium-sized datasets. Our method involves combining tree embedding with readily available nearest neighbor search packages. The potential scalability of our method relies on both the scalability of the tree embedding technique and the chosen nearest neighbor package.
>
> Regarding tree embedding, our current implementation effectively scales up to handling around $10^4$ documents. However, extending this scalability to accommodate millions to billions of documents poses challenges. We are actively exploring the incorporation of efficient clustering algorithms to achieve this scalability goal. Nevertheless, refining the weight estimation process for the tree methods remains an ongoing task.
>
> Additionally, we've encountered a limitation in the majority of nearest neighbor packages, as they lack support for sparse formats. Given that our tree embedding produces highly sparse vectors, addressing this limitation is crucial for handling larger datasets. It's worth noting that devising an efficient method for approximated nearest neighbor search with sparse data is a vibrant area of ongoing research.
>
> We will include these discussion in the final version.
>
> > (low priority) It would be nice to see performance of WMD on additional tasks besides knn classification. For example, to retrieve documents for the BEIR tasks (perhaps using the approach from Hyde) or when retrieving passages in open domain QA.
>
> Thank you very much for your suggestion. In the final version, we plan to include  nearest neighbor search experiments based on tree based methods and report top-1 and top-5 scores, where WMD is a ground truth.
>
> > What is speed performance when using a CPU faiss index? The GPU index may become untenable with larger data.
>
> The speed performance when using a CPU faiss index is presented in the following Table 3.
>
> Table 3 : Average testing time of the datasets used in the experiments for k $\in$ \{1,5,10\} using a CPU faiss index.
> |         |       | Twitter |       |      | BBCSport |       |      | Amazon |       |      | Classic |       |
> |:---:|:---:|:---:|:---:|:---:|:---:|:---:|:---:|:---:|:---:|:---:|:---:|:---:
> | Methods | 1     | 5       | 10    | 1     | 5        | 10    | 1     | 5      | 10    | 1     | 5       | 10    |
> | Faiss TWD |   3.969 |   3.917 |   4.033 |   0.389 |   0.347 |   0.351 |   174.105 |   169.923 |   165.540 |   81.241 |   80.908 |   81.603 |
> | Faiss ll-TWD $n\_{slice}=1$ |   0.430 |   0.428 |   0.449 |   0.044 |   0.044 |   0.045 |   14.749 |   15.265 |   15.228 |   10.330 |   9.860 |   9.997 |
> | Faiss ll-TWD $n\_{slice}=3$ |   2.224 |   2.319 |   2.336 |   0.156 |   0.159 |   0.139 |   47.057 | 47.426 |   47.380 |   30.582 |   30.345 |   30.283 |
> | Faiss ll-TWD $n\_{slice}=5$ |   4.205 |  4.331 |  4.298 |   0.325 |   0.307 |   0.308 |   78.731 |   80.303 |   80.349 |   51.749 |   51.749 |   51.580 |
> | Faiss ll-TWD $n\_{slice}=10$ |  9.171 |   8.849 |   9.135 |   0.790 |   0.778 |   0.809 |   163.348 |   160.211 |   159.874 |   102.815 |   102.325 | 102.379 |
>
> > Have you considered a continuous bag of words baseline? For example by averaging the word vectors then performing kNN? Kusner et al shows this does not work well, but it could still be interesting to sanity check and/or include as a row in the table to strengthen the motivation for using WMD.
>
> Thank you for the suggestions. No, we have not run the continuous bag of words baseline. During the rebuttal period, we quickly run the BOW experiments, and we found that the proposed method outperforms BOW.

---

### Meta-Review · Area_Chair_diug · 2023-09-19

**Recommendation:** 4

**Metareview:**

The authors introduce an efficient approximation to the computation of the Word mover's distance, a distance metric used to compare distributions which has found applications in NLP through its usage for document classification.
Reviewers highlighted the efficiency brought by the use of approximate nearest neighbour search while only resulting in minor performance drop when using the original formulation. The method seems readily applicable and could open up venues to the usage of Wasserstein distances to larger datasets. At the same time, concerns were raised on the limitations of using this to only one variation of that distance, as well as the small datasets used to perform experiments. However, this work fulfils the call for a short paper which is to propose one neat idea.

---

### Decision · Program_Chairs · 2023-10-07

**Decision:**

Accept-Main

**Comment:**

The authors introduce an efficient approximation to the computation of the Word mover's distance, a distance metric used to compare distributions which has found applications in NLP through its usage for document classification.
Reviewers highlighted the efficiency brought by the use of approximate nearest neighbour search while only resulting in minor performance drop when using the original formulation. The method seems readily applicable and could open up venues to the usage of Wasserstein distances to larger datasets. At the same time, concerns were raised on the limitations of using this to only one variation of that distance, as well as the small datasets used to perform experiments. However, this work fulfils the call for a short paper which is to propose one neat idea.